# A Facile HPLC-UV-Based Method for Determining the Concentration of the Bacterial Universal Signal Autoinducer-2 in Environmental Samples

**Kibaek Lee [1]**, **Chung-Hak Lee [2]** and **Kwang-Ho Choo [3,4,*]**

1 Department of Biotechnology and Bioengineering, Chonnam National University, Gwangju 61186, Korea; kibaek@jnu.ac.kr
2 School of Chemical and Biological Engineering, Seoul National University, Seoul 08826, Korea; leech@snu.ac.kr
3 Department of Environmental Engineering, Kyungpook National University, Daegu 41566, Korea
4 Advanced Institute of Water Industry, Kyungpook National University, Daegu 41566, Korea
* Correspondence: chookh@knu.ac.kr; Tel.: +82-53-950-7585

**Abstract:** As a universal quorum sensing (QS) signal, autoinducer-2 (AI-2) is utilized by both Gram-negative and Gram-positive bacteria to coordinate several group behaviors, such as biofilm formation, virulence, and motility, when the bacterial cell density exceeds the thresholds. The determination of the AI-2 level is essential to understand the physiological and biochemical processes involved in bacterial communication. However, the current methods for AI-2 determination are complicated, time-consuming, and require costly equipment, such as a mass spectrometer (MS) or fluorescence detector (FLD). In this study, we present a new and easily applicable method for AI-2 determination. This method, based on the primary derivatization of AI-2 with 2,3-diaminonaphthalene (DAN), uses an affordable high-performance liquid chromatography (HPLC) instrument with a UV detector. Under optimized conditions, our method showed a good linearity ($r^2 = 0.999$) and demonstrated the effective detection of AI-2 levels in various environmental samples, as follows: 0.38 ($\pm$0.05) $\mu$M for *E. coli* K12, 0.48 ($\pm$0.05) $\mu$M for *Aeromonas* sp. YB-2, 0.32 ($\pm$0.06) $\mu$M for the *Enterobacter* sp. YB-3, and 0.28 ($\pm$0.16) $\mu$M for activated sludge.

**Keywords:** autoinducer-2; (*S*)-4,5-dihydroxy-2,3-pentandione; HPLC-UV; quorum sensing; activated sludge

## 1. Introduction

Bacteria can sense the presence of, and communicate with, neighbors by detecting chemical signals called autoinducers. This phenomenon, called quorum sensing (QS), enables bacteria to maintain their ecological niche and form an environment favorable for their survival. QS involves the regulation of gene expression, such as biofilm formation, dispersion, conjugation, virulence, symbiosis, motility, and morphology, in response to variations in bacterial cell density [1–3]. Three types of QS signaling molecules are known to be involved in bacterial communication—Gram-negative bacteria use *N*-acylhomoserine lactones (AHLs or HSLs), while Gram-positive bacteria use autoinducer peptides (AIPs) or oligopeptides. In addition, autoinducer-2 (AI-2) is used as a universal signal molecule by both Gram-negative and Gram-positive bacteria [1,4]. Owing to its important implications in medical and environmental research, QS has been extensively studied for decades [5–10]. Accordingly, the development of methods to detect QS signals has been a key issue in the field of QS. Various methods, such as bioassay [11–14], high-performance liquid chromatography (HPLC) [15,16], liquid chromatography–mass spectrometry (LC–MS) [17,18], and gas chromatography–mass spectrometry (GC–MS) [19], have been developed for the analysis of AHLs, demonstrating that the detection of these molecules is relatively easy.

The detection of AI-2, however, has proven to be more complicated, as it can form various equilibrium structures through spontaneous cyclization in the aqueous phase (Figure S1) [20].

The current methods used to detect AI-2 can be classified into two categories (Table 1): (i) methods that require a derivative step (e.g., HPLC with fluorescence detector (HPLC-FLD) [21], LC–MS [22,23], GC–MS [24], etc.) and (ii) methods that do not require derivatization (e.g., AI-2 bioassay [25,26]).

**Table 1.** Methods of AI-2 detection reported in the literature.

| Methods | LOD (ng/mL) | Procedure | Instruments | Interference | References |
|---------|-------------|-----------|-------------|--------------|------------|
| HPLC-FLD [a] | 1.0 | Needs a derivative step, but is relatively easy to apply | Expensive | None | [21] |
| LC-MS/MS [a] | 0.7 | Derivatization reagents are complicated and time-consuming (to prepare) | Very expensive | Salt concentration or none | [22] |
| GC-MS [a] | 0.7 [c] | Requires complex sample pretreatment, including two-step derivatization and extraction | Very expensive | None | [24] |
| AI-2 bioassay [b] | 4.6 | Takes a long time to incubate the AI-2 reporter strain (*V. harveyi* BB170) (>12 h) and react (5–7 h) with the sample | Expensive | Poor reproducibility, depending on reporter strain and sample state | [25,26] |

[a] Requires one or two additional derivative steps for AI-2 detection. [b] Does not require a derivative step for AI-2 detection. [c] Estimated by a signal-to-noise ratio (S/N) of 5.

One example of the second type is the widely applied AI-2 bioassay method that uses the bioluminescent response of an AI-2 reporter strain (e.g., *Vibrio harveyi* BB strains) to measure the intensity of the AI-2 signal. Although the AI-2 bioassay has a relatively low detection limit, it requires a long preparation (>12 h) and measurement (5–7 h) time, in addition to sophisticated analytical skills. Moreover, the reproducibility of the method is relatively poor and depends on the AI-2 reporter strain and the state of the sample. Conversely, the first type of methods (HPLC-FLD, LC–MS, and GC–MS), which require a derivatization procedure for the fixation of the AI-2 signal structures, have the disadvantages that the additional derivative step is relatively less sensitive, and that advanced and costly detectors, such as a mass spectrometer (MS) or an FLD, are required. Nevertheless, these methods have the advantages of a short measuring time (<30 min), high reproducibility, and ease of application.

In view of the limitations of the current AI-2 determination methods, we developed a new method that uses a relatively accessible and low-cost HPLC-UV detector. Here, we demonstrate the validity of this new method for various environmental samples of pure and mixed bacterial cultures, such as *E. coli* K12 [9,27], *Aeromonas* sp. YB-2 [9], *Enterobacter* sp. YB-3 [9], and activated sludge.

## 2. Materials and Methods

### 2.1. Chemicals

The AI-2 precursor (*S*)-4,5-dihydroxy-2,3-pentanedione (DPD; MW 132.115 g/mol) was purchased from Omm Scientific Inc. (Dallas, TX, USA). DPD solutions with concentrations ranging from 0.3125 to 10.00 μM were used as the standards. 2,3-Diaminonaphthalene (DAN) was purchased from Alfa Aesar (Haverhill, MA, USA). The DAN solution was prepared by dissolving 0.2 mg DAN into 1.0 mL of 0.1 N HCl (Samchun, Korea). Acetonitrile (ACN) and formic acid (FA) were purchased from Sigma-Aldrich (Saint Louis, MO, USA). All of the solvents and other chemicals used were of analytical or HPLC grade.



## 2.2. Bacterial Strains and Culture Conditions

The bacteria used in this study were *E. coli* K12 [9,27], *Aeromonas* sp. YB-2 [9], and *Enterobacter* sp. YB-3 [9], which were isolated from activated sludge in a membrane bioreactor (MBR) for wastewater treatment. The bacteria were cultured in Luria–Bertani (LB) broth (Difco, USA) at 30 °C and 200 rpm in a shaking laboratory incubator.

## 2.3. Procedures of Sample Preparation and Derivatization for AI-2 Detection

When *E. coli* K12, *Aeromonas* sp. YB-2, *Enterobacter* sp. YB-3, and activated sludge cultures reached an optical density of 2.0–3.5 at 600 nm, each culture broth was centrifuged at $8000 \times g$ for 10 min at 4 °C, and then filtered through a 0.2-μm syringe filter (PVDF, Pall, New York, NY, USA) to remove the cells and debris. Samples of the DPD standard and cell-free supernatant (600 μL each) were transferred to 1.5 mL Eppendorf Safe-Lock Tubes (Eppendorf, Hamburg, Germany) containing an equal volume of DAN solution [21]. In the blank sample, an equal volume of deionized water was added instead. The two solutions were thoroughly mixed for 2 min and reacted for 40 min at 90 °C with linear shaking at 100 rpm in a water bath, were cooled in a refrigerator at 4 °C for 10 min, and then filtered through a 0.45 μm syringe filter (PVDF, Pall, USA) and immediately subjected to HPLC analysis. The data from triplicate measurements were averaged, and the standard deviations were calculated.

## 2.4. Chromatographic Procedures

Each 50 μL sample prepared in Section 2.3 was injected into an HPLC system equipped with a UV detector (Waters, USA) at a wavelength of 225 or 268 nm. The AI-2 in the injected samples was separated using a Phenomenex Luna 5 μm C18 reverse-phase column (150 × 2.0 mm). The mobile phase included 0.1% FA and pure ACN at a flow rate of 0.5 mL/min. The sequence of gradient elution was as follows: time (t) = 0 min, 70% for FA, 30% for ACN; t = 4 min, 70% for FA, 30% for ACN; t = 12 min, 35% for FA, 65% for ACN; t = 20 min, 35% for FA, 65% for ACN; t = 24 min, 70% for FA, 30% for ACN; and t = 27 min, 70% for FA, and 30% for ACN.

## 3. Results and Discussion

### 3.1. Optimal UV Wavelength for the Detection of the AI-2 Derivative

DPD was reacted with DAN to produce the AI-2 derivative (1-(3-methyl-benzo[g] quinoxaline-2-yl)-ethane-1,2-diol; Figure 1).

**Figure 1.** Product of the reaction of (*S*)-4,5-dihydroxy-2,3-pentandione (DPD) with 2,3-diaminonaphthalene (DAN).

The reaction mixture was eluted using an HPLC with a UV detector at a wavelength of 225 nm (Figure 2a). The second peak, appearing at a retention time of 1.5 min, corresponded to DAN. The third peak at a retention time of approximately 3 min corresponded to the derivative of DPD and DAN. The structure of the derivative shown in Figure 2a was confirmed by mass spectrometry (Figure S2). The eluted sample corresponding to the derivative was further scanned using a UV spectrometer in the range of 200–380 nm. The derivative showed specific absorption peaks at 225, 268, and 365 nm (Figure 2b). As the peak at 268 nm gave the highest intensity, it was adopted for the subsequent analysis of the derivative.

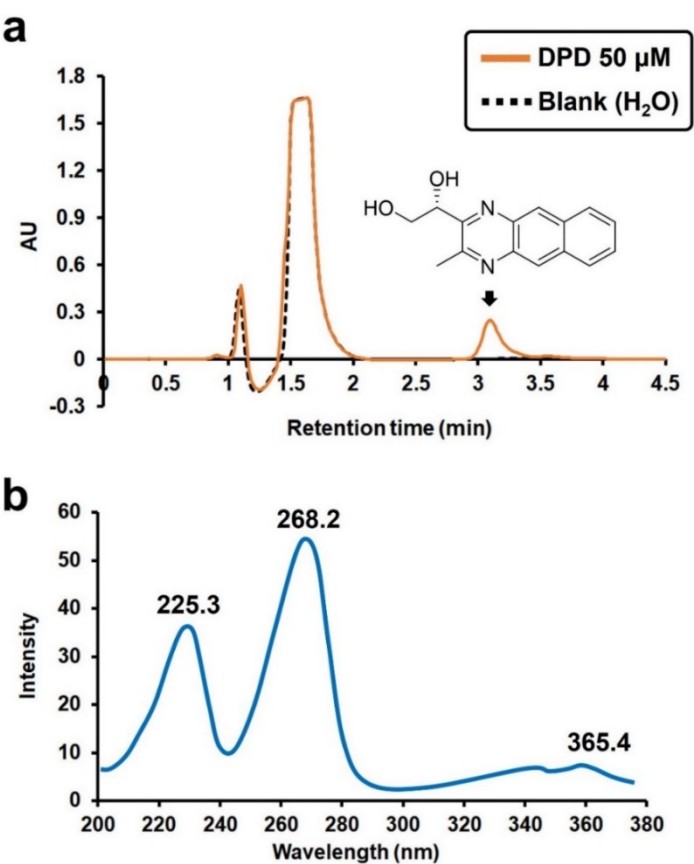

**Figure 2.** (**a**) HPLC chromatogram of the reaction mixture of DPD and DAN with a UV detector at 225 nm, and (**b**) UV spectrum for the AI-2 derivative.

### 3.2. *Validation of the Method for the Quantitative Analysis of DPD Concentration*

As the goal of this study was to quantitatively determine the DPD concentrations in various environmental samples, the DPD concentrations were controlled in a wide range of 0.3125–10.00 µM, equivalent to 41.30–1322 ng/mL in the constant excess concentration of DAN for the reaction in Figure 1. Subsequently, each reaction mixture was analyzed by HPLC to monitor each peak of the AI-2 derivative corresponding to each DPD concentration. The chromatogram was obtained using a UV detector at 268 nm (Figure 3a).

The plot of the peak area versus the DPD concentration showed good linearity ($r^2 = 0.999$), proving that this analytical method is precise enough to be applied to the quantitative measurement of DPD (Figure 3b). The limit of detection (LOD) for the HPLC-UV method developed in this study was determined to be 0.25 µM (33 ng/mL), using the linear-regression model [28]. This value is higher than those of the other methods listed in Table 1; however, the cost of equipment for other methods is very high (LC–MS/MS, GC–MS) and their reproducibility is poor (AI-2 bioassay). Although the reported LOD for HPLC-FLD is lower, in practice, there was no significant difference between the two methods. In addition, there was no particular inconvenience when using HPLC-UV for the detection of trace amounts of DPD in environmental samples, such as activated sludge in wastewater, as long as such a sample is concentrated, for instance by liquid–liquid extraction. This is discussed further in the following section.

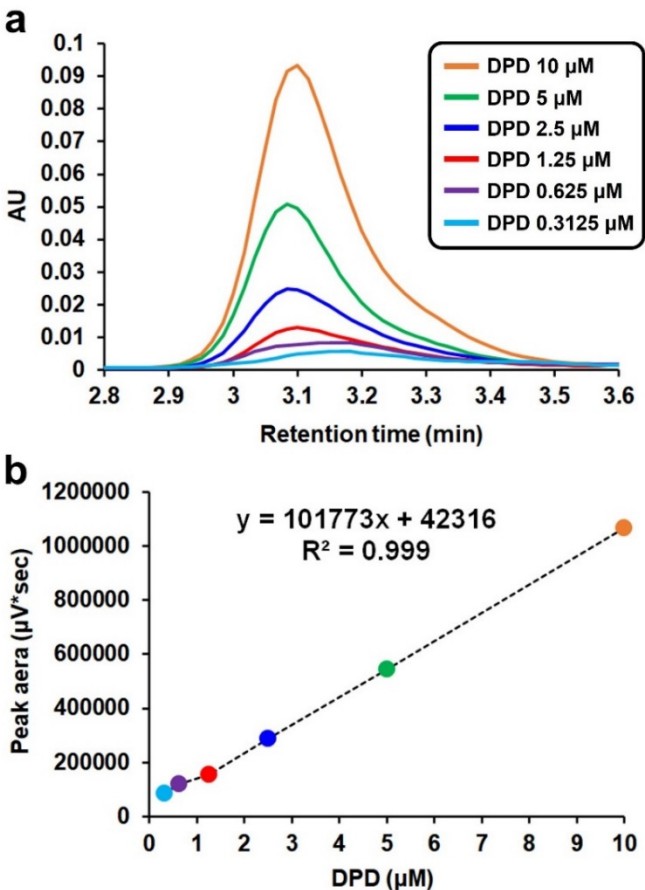

**Figure 3.** (**a**) Chromatograms of the AI-2 derivative (1-(3-methyl-benzo[g]quinoxaline-2-yl)-ethane-1,2-diol), a reaction product of DPD and DAN, as a function of DPD concentration (HPLC with a UV detector at 268 nm). (**b**) Plot of peak area versus DPD concentration.

### 3.3. Measurement of the DPD Concentration in Various Environmental Samples

To test whether the method developed in this study can be used to measure DPD concentrations in environmental samples, we selected environmental samples presumed to contain DPD: cell cultures, such as *E. coli* K12, *Aeromonas* sp. YB-2, *Enterobacter* sp. YB-3, and activated sludge in MBRs.

As shown in Figure 4, the DPD concentrations were 0.38 ($\pm$0.05) $\mu$M for K12, 0.48 ($\pm$0.05) $\mu$M for YB-2, and 0.32 ($\pm$0.06) $\mu$M for YB-3, showing relatively narrow standard deviations and thus a good precision. After culturing the microorganisms (K12, YB-2, YB-3, and activated sludge) for approximately 12 h, considerable amounts of DPD were found to have accumulated in the supernatant. In particular, the DPD concentration was 0.28 ($\pm$0.16) $\mu$M for the activated sludge in this study. In previous reports, (i) the DPD concentrations determined by HPLC-FLD for *E. coli* MG1655 and *V. harveyi* BB120 were 3.7 $\mu$M (486.1 ng/mL) and 35.8 $\mu$M (4725.6) ng/mL, respectively [29], and (ii) the AI-2 levels in the activated sludge in MBR liquor determined by LC–MS ranged from 0.0038–0.014 $\mu$M (0.5–1.8 ng/mL) [23].

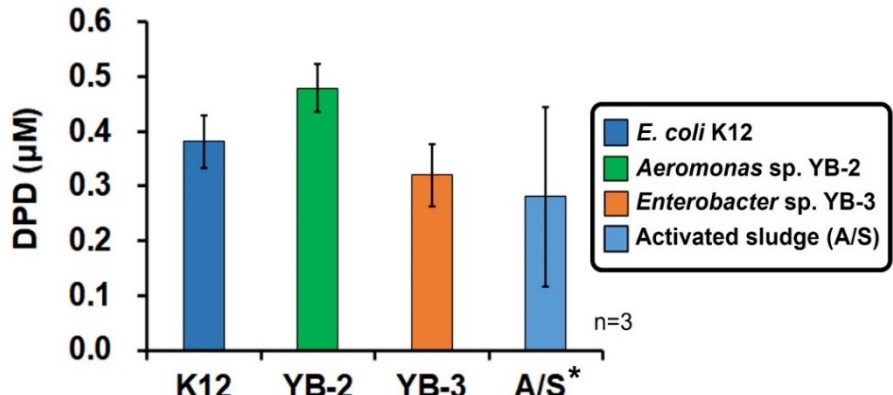

**Figure 4.** Determination of the (*S*)-4,5-dihydroxy-2,3-pentandione (DPD) concentration secreted from various microorganisms. The error bars represent one standard deviation ($n = 3$, * $n = 2$).

## 4. Conclusions

We developed a new analytical method for AI-2 (DPD) detection using the preparation of the DPD derivative, followed by HPLC analysis using a UV detector. The advantage of this method is that it utilizes a UV detector, which is more affordable than the other available methods for AI-2 detection (e.g., MS and FLD). Although the proposed method achieved a rather high LOD level compared with that of the previous methods, it demonstrated that it could be effectively used to measure AI-2 in bacterial cultures or environmental samples. Therefore, we suggest that this low-cost and readily accessible method is suitable for the quantification of AI-2 in a variety of environmental, medical, and other samples.

**Supplementary Materials:** The following are available online at https://www.mdpi.com/article/10.3390/app11199116/s1, Figure S1: Equilibrium form of (*S*)-4,5-dihydroxy-2,3-pentandione (DPD) and its derivatives in water and in the presence of borate, Figure S2: Total ion current (TIC) chromatogram obtained with LTQ XL Orbitrap high-resolution mass spectrometry with electrospray ionization (ESI) in a positive mode (Thermo Fisher Sci. Inc., Waltham, MA, USA).

**Author Contributions:** Methodology and writing (original draft preparation), K.L.; conceptualization and editing, C.-H.L.; writing (review and editing), K.-H.C. All authors have read and agreed to the published version of the manuscript.

**Funding:** This work was supported by the National Research Foundation of Korea (NRF-2021 R1C1C1008369).

**Institutional Review Board Statement:** Not applicable.

**Informed Consent Statement:** Not applicable.

**Data Availability Statement:** Not applicable.

**Acknowledgments:** ORCID, Kibaek Lee http://orcid.org/0000-0001-6877-1475, Kwang-Ho Choo http://orcid.org/0000-0002-4773-5886.

**Conflicts of Interest:** The authors declare no conflict of interest.

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
