# Peer review of "A Facile HPLC-UV-Based Method for Determining the Concentration of the Bacterial Universal Signal Autoinducer-2 in Environmental Samples"

_applsci, doi:10.3390/app11199116_

Round 1
Reviewer 1 Report
The authors have described a simple HPLC-UV method for detection and quantification of AI-2 in environmental samples. Although the method is easy in use, it has some disadvantages compared to existing chromatography-based methods employing either HPLC-FLD, LC-MS/MS or GC-MS. The established LOD of the HPLC-UV method is 33 to 47 times less sensitive than the LODs of the published methods. The question would be what sensitivity is required for reliable detection of AI-2 in bacterial cultures or environmental samples. As mentioned by the authors, the sensitivity of the HPLC-UV method could be improved by a sample pre-concentration step. How would this be done, how much work would this be, and what would be the concentration factor achievable? I also don’t think that a FLD detector is such a costly investment which only a few laboratories can afford. The authors state themselves in lines 155-156 “cost FLD slightly higher than UV”. One main question would also be: how specific is the reaction of the precursor with the chosen DAN compound? Finally, I noticed that the supplementary material is not included in the file.
Author Response
Response to Reviewer 1 Comments
"Please see the attachment"

Reviewer 2 Report
Abstract and keywords are excellent to describe e summarise the content of this manuscript.
The introduction should be improved about the state of art (add more references); authors should focus on the main research topics and relevant questions to be addressed. Some references should be added in relation to the proposed methods and relative technologies for the considered analysis (no previous studies?). Is it fully innovative?
Materials and methods are well described and detailed, but it will be interesting to add some information on calibration and validation data. Did authors use validated methods?
In the result/discussion section authors should better compare their results with other analytical and instrumental approaches used for similar purposes.
The conclusion is clear in relation to the study, but they should be linked in a better way to the other parts of the paper. Delete redundant information please.
Author Response
Response to Reviewer 2 Comments
"Please see the attachment"

Round 2
Reviewer 1 Report
Ad manuscript applsci-1405272v2
I agree with the explanations of the authors given in the response letter and the revised manuscript. I suggest the following two final changes in the text before acceptance of the manuscript:
Validation of the method
Line 160, … as long as such a sample is concentrated, for instance by liquid-liquid extraction.
Conclusions
Lines 186-188, Although … previous methods, it could be demonstrated that it can be effectively used to measure AI-2 in bacterial cultures or environmental samples.
Author Response
Response to Reviewer 1 Comments
Point 1: I agree with the explanations of the authors given in the response letter and the revised manuscript. I suggest the following two final changes in the text before acceptance of the manuscript:
Validation of the method, Line 160, … as long as such a sample is concentrated, for instance by liquid-liquid extraction.
Response: We revised it as you mentioned. Thank you.
Point 2: Conclusions, Lines 186-188, Although … previous methods, it could be demonstrated that it can be effectively used to measure AI-2 in bacterial cultures or environmental samples.
Response: We revised it as you mentioned. Thank you.
"Please see the attachment."
